# Expression of Semaphorin 3A in Malignant and Normal Bladder Tissue: Immunohistochemistry Staining and Morphometric Evaluation

**DOI:** 10.3390/biology10020109

**Published:** 2021-02-03

**Authors:** Ilan Bejar, Jacob Rubinstein, Jacob Bejar, Edmond Sabo, Hilla K Sheffer, Zaher Bahouth, Sarel Halachmi, Zahava Vadasz

**Affiliations:** 1The Department of Urology, Shamir Medical Center, Be’er Ya’akov 6093000, Israel; ilanbejar@gmail.com; 2The Department of Mathematics Technion, Israeli Institute of Technology, Haifa 3200003, Israel; koby@technion.ac.il; 3The Department of Pathology, Bnai Zion Medical Center, Haifa 31048, Israel; jacob.bejar@b-zion.org.il; 4The Department of Pathology, Carmel Medical Center, Haifa 34361, Israel; edmond.sabo@clalit.gov.il (E.S.); Hilla.sheffer@clalit.org.il (H.K.S.); 5The Department of Urology, Bnai Zion Medical Center, Haifa 31048, Israel; sarel.halachmi@b-zion.org.il; 6Proteomic Unit, Division of Clinical Immunology, Bnai Zion Medical Center, Haifa 31048, Israel; zahava.vadas@b-zion.org.il

**Keywords:** Sema3A, urothelial carcinoma of bladder, Immunohistochemistry

## Abstract

**Simple Summary:**

Semaphorin 3A (Sema3A) was shown to play a significant role in different neoplasms. In a previous study by our team, we showed that Sema3A is overexpressed in patients with urothelial carcinoma (UC). In this study, we analyzed 43 specimens from patients with the entire spectrum of UC and compared them with samples from 14 normal urothelium using immunostaining and computerized morphometry. The results showed that patients with UC had intense Sema3A staining in the apical layer of the mucosa compared to patients without UC. Moreover, patients with higher grade UC showed intense Sema3A staining across all mucosal layers.

**Abstract:**

Introduction: Our previous studies showed elevated levels of Semaphorin3a (Sema3A) in the urine of patients with urothelial cancer compared to healthy patients. The aim of this study was to analyze the extent of Sema3A expression in normal and malignant urothelial tissue using immune-staining microscopic and morphometric analysis. Materials and Methods: Fifty-seven paraffin-embedded bladder samples were retrieved from our pathology archive and analyzed: 14 samples of normal urothelium, 21 samples containing low-grade urothelial carcinoma, 13 samples of patients with high-grade urothelial carcinoma, 7 samples containing muscle invasive urothelial carcinoma, and 2 samples with pure urothelial carcinoma in situ. All samples were immunostained with anti Sema3A antibodies. The area of tissue stained with Sema3A and its intensity were analyzed using computerized morphometry and compared between the samples’ groups. Results: In normal bladder tissue, very light Sema3A staining was demonstrated on the mucosal basal layer and completely disappeared on the apical layer. In low-grade tumor samples, cells in the basal layer of the mucosa were also lightly stained with Sema3A, but Seama3A expression intensified upon moving apically, reaching its highest level on apical cells exfoliating to the urine. In high grade urothelial tumors, Seama3A staining was intense in the entire thickness of the mucosa. In samples containing carcinoma in situ, staining intensity was high and homogenous in all the neoplastic cells. Conclusions: Sema3A may be serve as a potential non-invasive marker of urothelial cancer.

## 1. Introduction

Urothelial carcinoma (UC) of the bladder includes a range of histologically and clinically diverse entities. UC is subcategorized into non-muscle invasive disease (including low-grade non-invasive disease, carcinoma in situ and high-grade non-muscle invasive disease) and muscle-invasive and metastatic disease [1,2].

Despite its common prevalence, there still is a lack of knowledge concerning the biological and molecular pathways leading to malignant transformation and disease progression. Treatment of non-muscle invasive disease and metastatic disease depends on Bacillus Calmette–Guérin (BCG) and chemotherapy, respectively, and a targeted effective medication has not been found yet, although recently Prembrolizumab, a highly specific, monoclonal antibody directed against PD-1, has demonstrated clinical efficacy in the treatment of advanced UC [3].

In our previous studies, we proved that Sema3A was overexpressed in urine samples of UC patients, whereas very low levels were found in urine samples of patients with no evidence of malignancy [4].

Semaphorins are a family of membrane-bound and soluble proteins classified into eight sub-classes based on their structural domains. Semaphorins regulate focal adhesion and cytoskeletal remodeling, thus affecting cell shape, cell attachment to the extracellular matrix, cell motility, and cell migration, all crucial characteristics essential for invasion and metastatic spread [5,6]. Although Semaphorins have been originally identified as having a role in axon guidance during development of the nervous system, they are now thought to fulfill diverse physiological roles including organogenesis, vascularization, angiogenesis, neuronal apoptosis, and neoplastic transformation. It was recently shown that Semaphorins were related to the regulation of the immune system, local cancer spread, metastases, cancer prognosis, and chemoresistance [7,8]. In particular, Sema3A was found to be play a role in several malignancies. For example, it was shown that Sema3A impedes tumor cell migration in breast cancer [9], and is also overexpressed in metastatic cells of patients with prostate cancer [10] and lung cancer [11].

Moreover, Semaphorins are also considered as a therapeutic target in few autoimmune diseases [12]. The role of Sema3A in UC was described in our previous study; however, there is a need for a better understanding of its role in the development and progression of UC.

The aim of this study was to demonstrate the ability of Sema3A to differentiate between UC and normal bladder tissue and between the different subtypes of UC by using immuno-staining of different bladder tissues.

## 2. Materials & Methods

### 2.1. Patients and Samples Collection

Fifty-seven formalin-fixed paraffin-embedded (FFPE) urinary bladder samples from patients with suspected bladder tumors were analyzed. 14 samples were from patients with no evidence of UC, 13 of them had a history of UC, and 1 with no history of UC. 11 out of these 13 patients have been previously treated with intravesical treatment (BCG or Mitomycin). All other 43 samples contained tumor cells with the following pathological results: 11 samples contained papillary urothelial carcinoma of low-malignant potential (PUNLMP), 7 samples contained non-invasive low-grade urothelial carcinoma (TaLGUC), 5 samples contained non-invasive high-grade urothelial carcinoma (TaHGUC), 3 samples contained lamina-invading low-grade urothelial carcinoma (T1LGUC), 8 samples contained lamina-invading high-grade urothelial carcinoma (T1HGUC), 7 samples contained muscle-invasive urothelial carcinoma (T2 HGUC), and 2 samples contained pure carcinoma in-situ (CIS) lesions. The clinical stage, histological type, and tumor grade were assessed using the World Health Organization 2016 guidelines. The mean age of the patients was 76 (37–94) years. Biopsies were numbered, diagnosed, and stored in the archive of pathology department at Bnai-Zion Medical Center, Haifa, Israel. The study was approved by the local ethical committee at Bnai-Zion Medical Center (Institutional Ethical Board) and signed patients’ consent were obtained from all study participants.

### 2.2. Tissue Processing

All the samples were fixed in 4% paraformaldehyde, processed and embedded in paraffin. Sections (3 μm) were mounted on SuperFrost slides (ThermoFisher Scientific; Waltham, MA, USA). Hematoxylin/Eosin staining was used for histological evaluation under light microscope. Sequential sections were used for Sema3A staining.

### 2.3. Immunohistochemistry

The immunostains were performed on an automated stainer (Benchmark Ultra; Ventana Systems, Phoenix, AZ, USA). Following antigen retrieval in Tris based buffer (32 min at 95–100 °C), Rabbit anti-Human Semaphorin 3A polyclonal antibody (NBP1-84409, Novus) (diluted 1:25) was used and incubated for 40 min. The detection reaction used the OptiView DAB detection kit (760–700 Ventana Systems, Phoenix, AZ, USA) according to manufacturer-recommended protocol. Hematoxylin counterstain was used for color development.

### 2.4. Expression of Sema3A Using Image Analysis

Images of sections mounted for Sema3A were taken using a DP70 Olympus camera. The epithelial layer was divided into three layers—basal, intermediate, and apical. The expression level of Sema3A was evaluated in each layer measuring its integrated optical density (IOD) values, using the image analysis software Image-Pro Plus (version 6.0, Media Cybernetics, Inc., Bethesda, MD, USA).

### 2.5. Statistical Analysis

The data was expressed as mean ± standard deviation (SD). Differences in the parameters were evaluated by *t*-test. A *p*-value of less than 0.05 was considered to be statistically significant.

## 3. Results

### 3.1. Immuno-Histochemical Analysis

We found a notable difference in Sema3A expression in each group, based primarily on the stage and grade of tissue differentiation. In normal bladder tissue, very light Sema3A staining was demonstrated on the basal layer of the mucosa. As cells migrate to the apical layers, stain intensity weakened, until it disappeared completely on the apical layer, where no Sema3A expression was found.

Sema3A expression followed a different pattern in lesions with a high level of cell differentiation (PUNLMP and noninvasive LGUC), a pattern somehow opposite to the one seen in normal bladder tissue; in these samples, cells in the basal layer of the mucosa were slightly stained (as the ones seen in normal tissue), but instead of intensity getting weaker with apical migration, Seama3A expression grew stronger and reached its highest level on the apical cells exfoliating to the urine.

In high grade urothelial carcinoma, Seama3A staining was intense in the entire thickness of the mucosa. These tumors lost the gradual change in stain intensity seen in normal tissue and low-grade UC.

In CIS samples, stain intensity was high and homogenous in all neoplastic cells (Figure 1).

Figure 1:1Normal bladder, Hematoxylin and eosin, ×402Normal bladder, Sema3A, ×40

Very light Sema3A staining is demonstrated on the basal layer of the mucosa.

As cells migrate to the apical layers, stain intensity gets even lighter, until it disappears completely on the apical layer.

3Low grade papillary urothelial carcinoma Ta, Hematoxylin and eosin, ×1004Low grade papillary urothelial carcinoma Ta, Sema3A, ×100

Cells in the basal layer of the mucosa show light Sema3A staining. Stain grows stronger as cells migrate versus the bladder lumen and reaches its peak on the apical cells exfoliating to the urine. 

5High grade urothelial carcinoma Ta, Hematoxylin and eosin, ×2006High grade urothelial carcinoma Ta, Sema3A, ×200

Sama3A staining is intense in the entire thickness of the mucosa without any change in stain intensity between the different urothelial layers.

7Carcinoma in situ, Hematoxylin and eosin, ×2008Carcinoma in situ, Sema3A, ×200

### 3.2. Morphometric Analysis

The urothelial mucosa was roughly divided into three layers: basal, intermediate, and apical. We examined the expression pattern of Sema3A by image analysis using IOD values. Results showed high expression of Sema3A in the basal (1506.3 ± 914.7), intermediate (1768.1 ± 917.3), and apical (1909.1 ± 961.8) layers of HGUC. It should be emphasized that the standard deviation in this group was high. However, in spite of the great variance between the HGUC samples, the expression of Sema3A was found to be significantly high compared to both LGUC and normal bladder mucosa in all layers. LGUC showed significantly high expression of Sema3A compared to normal bladder mucosa in the apical (653.1 ± 243.6 vs. 32.6 ± 14.1) and intermediate (237.6 ± 89.2 vs. 69.9 ± 38.0) layers (Figure 2).

## 4. Discussion

Our previous studies showed, for the first time, that Sema3A levels in the urine correlated well with the presence of UC. We also demonstrated a significant correlation between Sema3A levels and tumor size and the number of lesions. Moreover, Sema3A levels in healthy volunteers were significantly lower compared to the levels in patients with UC [4]. Following these observations, we tried to better understand the role of Sema3A in UC development and to give an explanation for the different levels of Sema3A in urine in different patients.

The results of the current study clearly shows that normal tissue has very low expression of Sema3A, which is limited mainly to the basal layer. UC shows a higher expression of Sema3A, and it correlates well with tumor stage and grade. Tumors with higher grade or stage showed a higher expression of Sema3A through all the mucosal layers.

These results prove the findings of our previous studies: faint staining in normal tissue correlates directly with lower levels of Sema3A in the urine of patients with no UC. Moreover, patients with higher levels of Sema3A in the urine are more likely to have higher grade and higher stage UC.

The current study along with our previous study provide evidence for the involvement of Sema3A in UC; however, we still do not have a direct understanding of the mechanism in which it is related to the malignant transformation. The paucity of Sema3A staining in normal tissue and the gradual increment in the area and intensity of the staining as cancer transforms to more aggressive forms clearly indicates that Sema3A has a crucial role in this transformation.

It is worth noting that most normal bladder samples were taken from patients previously treated with either BCG or Mitomycin. These patients showed a light expression of Sema3A in their bladder mucosa, similar to patients with no history of UC, despite the fact that their bladder tumor samples prior to therapy highly expressed Sema3A. Therefore, Sema3A could also help in the follow-up of patients following intravesical therapy as a non-invasive marker for tumor recurrence.

Sema3A promotes malignancy via inhibition of the immune system. Tumor cells present Sema3A and inhibits immune anti neoplastic response by blocking T cells adhesion capabilities, proliferation, and cytokine production [13]. The complex of Sema3A and neuropilin1 showed its capabilities to suppress the activity of tumor-associated macrophages by inhibiting their migration capabilities and shifting their phenotype from pro-immune and anti-tumoral activities [14].

Several studies showed that Semaphorin 3 class is related to malignant transformation in various tissues. Specifically, attempts have been made to decipher the role played by Sema3A in tumor growth and propagation. A number of studies showed that low levels of Sema3A were associated with non-small cell lung carcinoma [15] and melanoma [16]. Hu et al. [17] demonstrated up-regulation of Sema3A in metastatic cell lines, and from patients with tumor recurrence, leading to the conclusion that Sema3A promoted cell proliferation, cell migration, and invasion; moreover, survival analysis indicated that Sema3A can be an independent predictor of survival, and potential target for treatment [8].

Jiang et al. hypothesized that Sema3A is a potential tumor suppressor gene due to the fact that it is down regulated in numerous types of malignancies, including prostate, breast, ovarian and glioma [18].

The main limitations of our study are the relatively small number of patients in each group, and the fact that most normal bladder tissues were obtained from patients with a history of UC. However, these limitations do not affect the results and the conclusion in a significant way.

In summary, our two previous studies [4,19], along with this study, proved the association between Sema3A and the development and progression of UC, in addition to its valuable use as a noninvasive marker for cancer detection and follow up. Our data adds further evidence for the role of Semaphorins in malignant transformation in various tissues.

## 5. Conclusions

Normal urothelial tissue has low expression of Sema3A. Neoplastic tissues expresses higher levels of Sema3A in correlation with tumor stage and grade. Sema3A may be a potential noninvasive marker for diagnosing UC.

## Figures and Tables

**Figure 1 biology-10-00109-f001:**
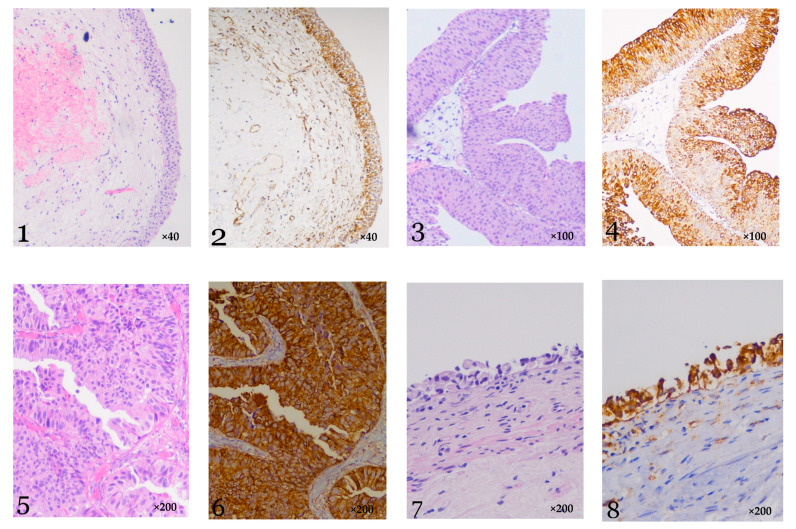
Hematoxylin & Eosin staining vs. Sema 3A immunohistochemical staining of: Normal bladder (**1**,**2**), Low grade papillary urothelial carcinoma (**3**,**4**), High grade urothelial carcinoma (**5**,**6**), and Carcinoma in situ (**7**,**8**).

**Figure 2 biology-10-00109-f002:**
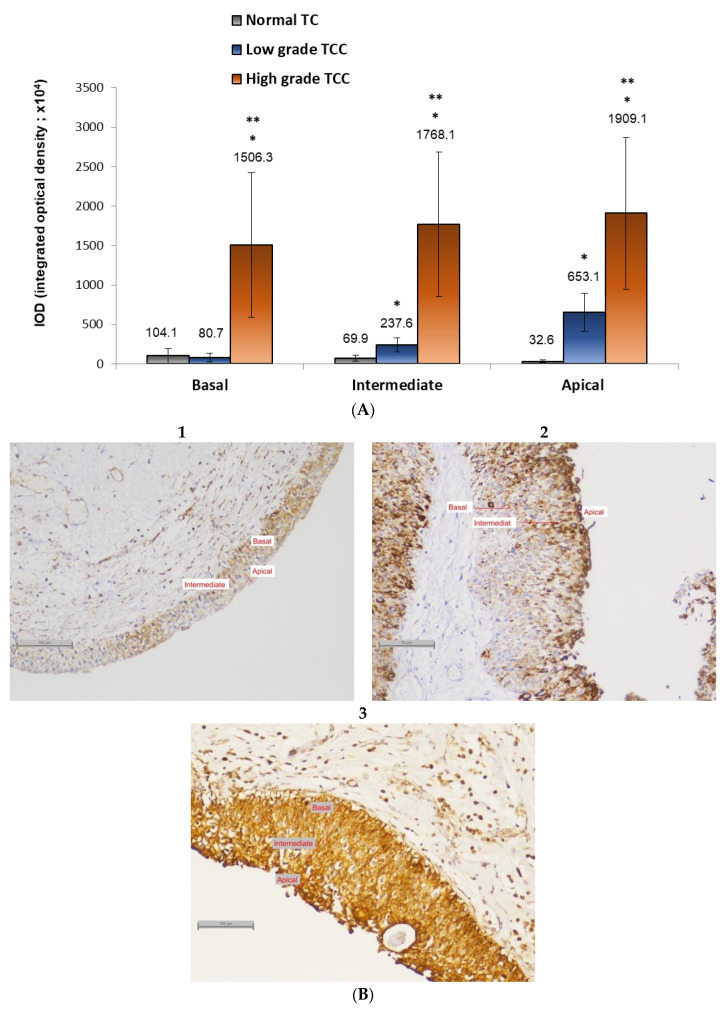
Expression of semaphorin 3A in UC. Immunohistochemical staining of high grade, low grade, and normal TC samples were analyzed for semaphorin 3A expression in basal, intermediate and apical layers of epithelium. Image analysis of repetitive pictures of epithelium sections was conducted (**A**). Three of the pictures are exhibited in (**B**) (1—Normal; 2–Low grade UC; 3—High-grade UC). Asterisks mark statistical significance (*p* < 0.05): * Performance vs. Normal; ** Performance vs. Low grade UC (×100).

## Data Availability

The data presented in this study are available on request from the corresponding author. The data are not publicly available due to patient privacy.

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
