# Peer review of "Expression of Semaphorin 3A in Malignant and Normal Bladder Tissue: Immunohistochemistry Staining and Morphometric Evaluation"

_biology, 2021, doi:10.3390/biology10020109_

Round 1

Reviewer 1 Report

This paper looks at protein expression (Sema3A) in UCC versus normal tissue and detects a tangible difference between the two. Moreover, there is a difference in pattern of expression from LG to HG and CIS. This is an interesting finding. I would have someone fluent in English read the paper to fix the minor mistakes spotted in the paper. Beside that, use the newer terminology (UCC instead of TCC). Figure 2 needs to be color coded so the reader can understand what each column presents. Otherwise, this is a nicely done paper.

Author Response

Point 1: I would have someone fluent in English read the paper to fix the minor mistakes spotted in the paper

Answer 1: revisions were made to fix minor language flaws.

Point 2: use the newer terminology (UCC instead of TCC)

Answer 2: thank you so much for your comment, this was corrected

Point 3: Figure 2 needs to be color coded so the reader can understand what each column presents

Answer 3: The figure is now color-coded

Reviewer 2 Report

1) Well written article. Conclusions are supported by the evidence provided.

2) Please expand BCG in the introduction.

3) It would be helpful for the readers if FIG 2 image is annotated with demarcations for the three layers: basal, intermediate and apical.

Author Response

Point 1: Well written article. Conclusions are supported by the evidence provided.

Answer: thank you so much

Point 2: Please expand BCG in the introduction.

Answer: thank you for the comment. done!

Point 3: It would be helpful for the readers if FIG 2 image is annotated with demarcations for the three layers: basal, intermediate and apical.

Answer: excellent point. Done!

Reviewer 3 Report

The manuscript is written correctly, the experiments are adequate and the results are discussed. However, both the introduction and the discussion lack recent references on the role of Semaphorin 3A. In particular, the controversial Semaphorin role should be discussed, perhaps discussing recent publications e.g. such as the following references:

Pham HT, Kondo S, Endo K, Wakisaka N, Aoki Y, Nakanishi Y, Kase K, Mizokami
H, Kano M, Ueno T, Hatano M, Moriyama-Kita M, Sugimoto H, Yoshizaki T.
Influences of Semaphorin 3A Expression on Clinicopathological Features, Human Papillomavirus Status, and Prognosis in Oropharyngeal Carcinoma. Microorganisms. 2020 Aug 22;8(9):1286. 

 Izycka N, Sterzynska K, Januchowski R, Nowak-Markwitz E. Semaphorin 3A
(SEMA3A), protocadherin 9 (PCdh9), and S100 calcium binding protein A3 (S100A3)as potential biomarkers of carcinogenesis and chemoresistance of differentneoplasms, including ovarian cancer - review of literature. Ginekol Pol.
2019;90(4):223-227. 

Author Response

Point 1: The manuscript is written correctly, the experiments are adequate and the results are discussed. However, both the introduction and the discussion lack recent references on the role of Semaphorin 3A. In particular, the controversial Semaphorin role should be discussed, perhaps discussing recent publications e.g. such as the following references:

Answer: We appreciate this comment. we have updated with relevant recent publications